# Approximating Naive Bayes on Unlabelled Categorical Data

**Cormac Herley**
*Microsoft Research*
*Redmond, WA*

**Reviewed on OpenReview:** *https://openreview.net/forum?id=KpElM2S9pw*

## Abstract

We address the question of binary classification when no labels are available and the input features are categorical. The lack of labels means supervised approaches can't be used, and the lack of a natural distance measure means that most unsupervised methods do poorly. For such problems, where the alternatives might be a) do nothing or b) heuristic rules-based approaches, we offer a third alternative: a classifier that approximates Naive Bayes. Our primary scenarios are those that involve distinguishing scripted, or bot, web traffic from that of legitimate users.

Our main assumption is the existence of some attribute $x_*$ more prevalent in the benign than the scripted traffic; i.e., $P(x_*|\overline{\text{bot}}) = K \cdot P(x_*|\text{bot})$, for $K > 1$. We show that any such disparity yields a lower bound on $P(\text{bot}|x_j)$ even when we have no prior estimates of $P(x_*|\overline{\text{bot}})$, $P(x_*|\text{bot})$ or $K$ (except that $K > 1$). We show that when at least one bin of at least one feature receives no attack traffic then we under-estimate the actual conditional probability by a factor of $1 - 1/K$. Thus, any attribute with a large disparity between prevalence in benign and abuse traffic (i.e., $K$ is large), allows good approximation of the Naive Bayes classifier without the benefit of labels.

The approach is particularly suited to problems where $K$ is high and thus the approximation is very accurate. Example problems (and relevant attributes) might be: password-guessing, if login attempts from legitimate users succeed at a much higher rate than those from password-guessing attackers; Credit Card Verification Value (CVV) guessing, if an attacker exhaustively tries all possible 3 or 4-digit values and fails at a higher rate than legitimate users; account registration, if legitimate users use email addresses from services that do not allow fee anonymous accounts (e.g., `.edu`) at a much higher rate than attackers; click-fraud if legitimate users visit pages and services that contain no ads at a higher rate than click-fraud bots.

## 1 Introduction

Abuse of web services often involves automated traffic. This includes registration abuse (i.e., scripting the account creation process), unauthorized scraping, click-fraud, password guessing, fake engagement on social networking sites and so on. This traffic can be hard to detect. Supervised approaches can't be used since labels are almost never available (Chio & Freeman, 2018). Anomaly detection and one-class learners that assume outliers are "few and rare" do not work well when abuse can range from 5% to 90% of traffic (Chandola et al., 2009). Most unsupervised methods assume that closeness in feature space implies greater likelihood of belonging to the same class. Since we have categorical features there is no natural idea of distance between samples; thus, unsupervised approaches like K means, One-Class Support Vector Machines (SVMs) (Schölkopf et al., 1999), and Isolation Forests (Liu et al., 2008) are not easily applied. Techniques that map categorical features to numeric (such as one-hot encoding) offer limited help when we have a small number of features (Zheng & Casari, 2018).

The scale of these abuse problems can be considerable. Twitter suspended 70 million fake accounts in June 2018. In April 2021, the information of 533 million Facebook and 500 million LinkedIn users were gathered by

unauthorized scraping. Akamai reports that 43% of login attempts on their platform in 2017 were password-guessing attempts (Akamai). Varol et al. (Varol et al., 2017) estimate that at least 15% of Twitter's active accounts are bot-created fakes.

In this paper we show that if we can identify a binary feature $x_*$ which is more prevalent in the benign traffic than in the abuse we can approximate a Naive Bayes (NB) classifier. That is, if $P(x_*|\overline{\text{bot}}) = K \cdot P(x_*|\text{bot})$, for $K > 1$ we can approximate $P(\text{bot}|x_j)$ in any feature $x_j$. The approximation is order-preserving: sorting a list of samples by our estimates will have the same order as the true NB estimates. Further, when $K$ is large, the approximation is very accurate.

Examples of possible problems where the technique might be worth considering are: password-guessing, if login attempts from legitimate users succeed at a much higher rate than those from password-guessing attackers; account registration, if legitimate users use email addresses from services that do not offer anonymous free accounts at a much higher rate than attackers; social networking, if the fraction of legitimate users for whom the follower/followed ratio is large to greatly exceed that for bot accounts.

For web-services the fields available will be restricted to those gathered from a browser. A very common list of features will be those listed in the Apache web-server Combined Log Format (CLF) (Apache, 2019). An example logged request might be:

```
127.0.0.1 [10/Oct/2000:13:55:36 -0700] "GET /apache_pb.gif HTTP/1.0" 200 2326
"http://www.example.com/start.html" "Mozilla/4.08 [en] (Win98; I ;Nav)"
```

These recorded features are: Client IP address, Timestamp, Request, HTTP response, Bytes Requested, Referer, Useragent. Note that, with the exception of Bytes Requested, all of the features are categorical or ordinal. Further note that we effectively have five or six features.

## 2  Related work

### 2.1  Unsupervised and weakly supervised learning

Most unsupervised learning methods assume that points that are close in feature space are more likely to belong to the same class (Hastie et al., 2001). This is true of clustering methods (e.g., $K$ means), Principal Components Analysis, Isolation Forests (Liu et al., 2008) and One-Class SVMs (Schölkopf et al., 1999). Similarly, a survey of anomaly detection approaches divides them into classification (i.e., supervised), nearest-neighbor, clustering and statistically based methods (Chandola et al., 2009); all of these require a reliable measure of distance. When we deal with categorical inputs some modification of the input space must be performed to generate numeric features.

A common way of accomplishing this is with one-hot encoding, where $d$ features $x_1, x_2, \cdots, x_d$ of cardinalities $R_1, R_2, \cdots, R_d$ will be mapped into numeric features each of which take the value zero or one. This mapping now allows us to use the unsupervised and anomaly detection techniques mentioned above. However, the distance in the new feature space is very constrained: the distance between any two points must be one of the discrete values $0, 1, \cdots, d$. Unfortunately, this severely limits the efficacy of distance-based unsupervised algorithms. This is particularly true when $d$ is small (as is the case in our web-services scenario where $d \approx 5$). It is easy to see that performance of an algorithm like $K$ means will not perform well when there are only $d + 1 = 6$ possible distances between points. A further difficulty in applying One-Class SVMs is that the algorithm requires a parameter $\nu$ which represents the fraction of samples that are expected to be anomalous. In anomaly detection applications we might expect this to be very small (e.g., $\nu < 0.01$) but in fraud and abuse problems we expect it to be much higher (e.g., $0.05 < \nu < 0.9$). In addition, the amount of fraud might vary quite rapidly as attackers gain or lose access to resources or alter their strategies. The need to provide the One-Class SVM algorithm with a prior is a significant shortcoming of the approach.

Our work belongs to the growing body of work on weak supervision. While most of the work on weak supervision involves noisy or unreliable labels, there are interesting variations. Parisi et al. show how to rank several different classifiers without the benefit of labels (Parisi et al., 2014). A similar weak supervision signal to ours is identified by Soman et al. (Soman et al., 2022). They are also interested in the classification

problem and identify markers that are more prevalent in one class than the other. They use this to evaluate the relative performance of competing models rather than to build a NB approximation as we do. Joyce et al. (Joyce et al., 2021) describe how some performance metrics for clustering and multi-class classification algorithms can be computed in the absence of ground-truth reference labels. The approach leverages prior domain knowledge and works best when the number of classes is large and grouping similar data points is easier than determining ground-truth labels. An ingenious approach to classification without labels is described in (Du Plessis et al., 2013) and developed further in (Kaji & Sugiyama, 2019). They show how the classification boundary between $C_0$ and $C_1$ may be obtained if we have two unlabelled collections of unknown mixtures of the two classes. While it builds a classifier this approach does not estimate the class membership probabilities $P(C_0|x)$ and $P(C_0|x)$. Hence (unlike our method) a ranking of samples from most to least bot-like isn't available; this can be problematic for abuse applications where sensitivity to false positives often means that we act only when $P(\text{bot}|x)$ is very high.

## 2.2  Detection of automated abuse

Despite the scale and importance of automated abuse it is unclear that Machine Learning (ML) has had significant impact. In fact, Sommer and Paxson describe the limitations of ML methods for security problems (Sommer & Paxson, 2010) observing that the absence of labels and available datasets are serious impediments. A more recent account of using ML to protect consumer properties is by Chio and Freeman (Chio & Freeman, 2018). Many of these problems are inherently unsupervised. This is not simply a question of cost or resources; even expert human labellers will have no basis to label individual web requests as malicious or benign. In fact, many abuse problems appear still addressed using hand-crafted rules and heuristics. They involve categorical variables, making direct use of one-class algorithms such as One-class SVM's (Schölkopf et al., 1999) and Isolation Forests (Liu et al., 2008) difficult.

IP reputation has often been suggested as a solution. When abuse traffic comes from a small IP space blocklist solutions might capture much of the abuse with minimal impact to the benign traffic (Stringhini et al., 2015). Unfortunately, in a recent large-scale honeypot study Li et al. find that 64.37% of bot IP addresses lie in the residential IP space (Li et al., 2021). It's obviously not feasible for web-scale consumer platforms such as Facebook, Google, Amazon, Pinterest, Twitter etc to block huge portions of the consumer IP space. CAPTCHA's (Ahn et al., 2003) to tell humans and computers apart have been widely deployed but suffer from a number of difficulties. They place a considerable burden on users and have been successfully broken by automated solvers (Bursztein et al., 2014).

Kudugunta and Ferrara (Kudugunta & Ferrara, 2018) describe a DNN approach to social bot detection. They use synthetic minority oversampling (Chawla et al., 2002) which generates a large amount of synthetic labelled data from a smaller number of actual labels. Jan et al (Jan et al., 2020) also leverage a small amount of labelled data to synthesize quantities that can be used by neural networks. They echo some of findings of Sommer and Paxson (Sommer & Paxson, 2010) on the limitations of ML methods. When pairs of features that are independent in the benign traffic can be identified, deviations from rank-one in the associated pivot matrices can be used to classify abuse traffic (Herley, 2022).

# 3  Target Encoding: Replace tokens with conditional probabilities

We assume that we receive an unknown mixture of benign and malicious traffic:

$$P(\boldsymbol{x}) = [1 - P(\text{bot})] \cdot P(\boldsymbol{x}|\overline{\text{bot}}) + P(\text{bot}) \cdot P(\boldsymbol{x}|\text{bot}), \tag{1}$$

where $\boldsymbol{x}$ is our vector of observed features.

We wish to build a classifier that distinguishes $\overline{\text{bot}}$ from bot traffic. An initial difficulty is that many ML techniques take numeric features as input, and thus categorical features must first be transformed. We discussed the significant problems of approaches such as one-hot encoding in Section 2 (Bilenko, 2015; Zheng & Casari, 2018).

In supervised settings one way around this problem is to replace each category token with the conditional probability of the target class under the token value. That is (if $x_{jk}$ is the $k$-th token of the $j$-th feature of

$\boldsymbol{x}$) replace $x_{jk}$ with $P(\text{bot}|x_j = x_{jk})$. When we have labels this can be done by simply counting the number of positive and negative instances ($N^+$ and $N^-$ respectively) in the $x_{jk}$ bin over a training window:

$$P(\text{bot}|x_j = x_{jk}) \approx \frac{N^+}{N^+ + N^-}.$$

When we apply to many features this is, of course, a component of the Naive Bayes classifier: the conditional probability assuming per-class independence of features. The trick of using this as input to a more sophisticated classifier appears to have been discovered and rediscovered many times (Chen et al., 2009; Li et al., 2010; Lee et al., 1998). Good treatments are given by Bilenko (Bilenko, 2015) and by Zheng and Casari (Zheng & Casari, 2018). Zheng and Casari point out that this transformation of the categorical variables has several advantages over one-hot encoding, such as being computationally more efficient, scaling to higher cardinality features, and working well with non-linear models.

Our first observation is that labels entered into the calculation of the conditional probability only via their aggregates $N^+$ and $N^-$. That is, if we are content with a Naive Bayes classifier we don't need the individual labels, just a way to estimate the ratio of the sum of their counts. If we had a mechanism to estimate $P(\text{bot}|x_{jk})$ or $N^+/(N^+ + N^-)$ directly we would not need labels at all.

## 4    Lower bound on conditional probabilities

Suppose we have a binary feature, $x_*$, such that $P(x_*|\overline{\text{bot}}) > P(x_*|\text{bot})$; that is, $x_*$ has value `True` at a higher rate in the benign than in the abuse traffic (although we don't know $P(x_*|\overline{\text{bot}})$ or $P(x_*|\text{bot})$).[1] Consider the conditional probability of $x_*$ conditioned on $x_{jk}$. From (1), we get:

$$P(x_*|x_{jk}) = [1 - P(\text{bot}|x_{jk})] \cdot P(x_*|\overline{\text{bot}}, x_{jk}) + P(\text{bot}|x_{jk}) \cdot P(x_*|\text{bot}, x_{jk}).$$

Hence:

$$P(\text{bot}|x_{jk}) = \frac{P(x_*|\overline{\text{bot}}, x_{jk}) - P(x_*|x_{jk})}{P(x_*|\overline{\text{bot}}, x_{jk}) - P(x_*|\text{bot}, x_{jk})}.$$

If we had labels we could estimate this directly, but without them we cannot. That is, while we can estimate $P(x_*|x_{jk})$ by counting positive and negative instances of $x_*$, we don't have a way of unravelling the values for $P(x_*|\overline{\text{bot}}, x_{jk})$ and $P(x_*|\text{bot}, x_{jk})$.

However, since we are interested in the Naive Bayes estimate we can simplify by imposing per-class independence among features. This gives $P(x_*|\overline{\text{bot}}, x_{jk}) = P(x_*|\overline{\text{bot}})$ and $P(x_*|\text{bot}, x_{jk}) = P(x_*|\text{bot})$, and hence:

$$P(x_*|x_{jk}) = [1 - P(\text{bot}|x_{jk})] \cdot P(x_*|\overline{\text{bot}}) + P(\text{bot}|x_{jk}) \cdot P(x_*|\text{bot}). \tag{2}$$

This simplifies our expression for the conditional probability to

$$P(\text{bot}|x_{jk}) = \frac{P(x_*|\overline{\text{bot}}) - P(x_*|x_{jk})}{P(x_*|\overline{\text{bot}}) - P(x_*|\text{bot})}. \tag{3}$$

Also, observe that (2) says that $P(x_*|x_{jk})$ is a probability-weighted sum of $P(x_*|\overline{\text{bot}})$ and $P(x_*|\text{bot})$; so it must lie between them. Since we know that $P(x_*|\overline{\text{bot}}) > P(x_*|\text{bot})$ we get the following ordering:

$$P(x_*|\overline{\text{bot}}) \geq P(x_*|x_{jk}) \geq P(x_*|\text{bot}). \tag{4}$$

We wish to be conservative in flagging traffic as malicious; hence we prefer to under- rather than over-estimate $P(\text{bot}|x_{jk})$. Thus, our main interest is in a lower bound. Obviously, we can get a lower bound on (3) by decreasing the numerator and/or increasing the denominator. Consider the denominator first. Since (4) implies $P(x_*|\overline{\text{bot}}) > P(x_*|\overline{\text{bot}}) - P(x_*|\text{bot})$ we get:

$$P(\text{bot}|x_{jk}) \geq \frac{P(x_*|\overline{\text{bot}}) - P(x_*|x_{jk})}{P(x_*|\overline{\text{bot}})}.$$

---

[1]While $x_*$ is an input feature we denote it separately from the other $d$ input features; i.e., $x_* \notin \{x_1, x_2, \cdots, x_d\}$.

This still leaves us with $P(x_*|\overline{\text{bot}})$ to estimate or bound. Again we invoke (4): the observed distribution $P(x_*|x_{jk})$ is lower than or equal to $P(x_*|\overline{\text{bot}})$, never higher. Thus $\max_{j,k} P(x_*|x_{jk})$ is a lower bound for $P(x_*|\overline{\text{bot}})$. So we define:

$$\hat{P}(x_*|\overline{\text{bot}}) \triangleq \max_{j,k} P(x_*|x_{jk}) \le P(x_*|\overline{\text{bot}}). \tag{5}$$

Here the maximization is over all bins of all features. This gives our overall lower bound for $P(\text{bot}|x_{jk})$:

$$\hat{P}(\text{bot}|x_{jk}) \triangleq 1 - \frac{P(x_*|x_{jk})}{\hat{P}(x_*|\overline{\text{bot}})} \le P(\text{bot}|x_{jk}). \tag{6}$$

That is, the left-hand side gives us a lower bound for the Naive Bayes estimate in terms only of quantities we observe.

A similar analysis gives a lower bound on the fraction traffic that is from bots:

$$\hat{P}(\text{bot}) \triangleq 1 - \frac{P(x_*)}{\hat{P}(x_*|\overline{\text{bot}})} \le P(\text{bot}). \tag{7}$$

To build our overall classifier, observe that:

$$
\begin{aligned}
P(\text{bot}|\boldsymbol{x}) &= \frac{P(\text{bot})}{P(\boldsymbol{x})} \cdot P(\boldsymbol{x}|\text{bot}) \\
&= \frac{P(\text{bot})}{P(\boldsymbol{x})} \cdot \prod_{i=1}^{d} \frac{P(\text{bot}|x_i) \cdot P(x_i)}{P(\text{bot})}.
\end{aligned}
$$

Replacing $P(\text{bot})$ and $P(\text{bot}|x_i)$ with their estimates, we get:

$$\hat{P}(\text{bot}|\boldsymbol{x}) \triangleq \frac{\hat{P}(\text{bot})}{P(\boldsymbol{x})} \cdot \prod_{i=1}^{d} \frac{\hat{P}(\text{bot}|x_i) \cdot P(x_i)}{\hat{P}(\text{bot})}.$$

The NB estimator decides for bot if $\hat{P}(\text{bot}|\boldsymbol{x}) > \hat{P}(\overline{\text{bot}}|\boldsymbol{x})$, and for $\overline{\text{bot}}$ otherwise. If we wish to trade true and false positives at some specific rate a threshold is easily incorporated in the decision.

It is worth noting, from (2), that if at least one bin of at least one feature is unattacked (i.e., $P(\text{bot}|x_{jk}) = 0$ for some $x_{jk}$) then $\hat{P}(x_*|\overline{\text{bot}}) = P(x_*|\overline{\text{bot}})$. This is a reasonable expectation unless abuse is spread to all bins of all features. For example, if no abuse comes from a less-common browser like Opera, or a geographic region like Wisconsin, then our estimate of $P(x_*|\overline{\text{bot}})$ in (5) is exact.

## 4.1   Tightness of the bound

We assumed that $P(x_*|\overline{\text{bot}}) > P(x_*|\text{bot})$. Let's call their ratio $K$:

$$P(x_*|\overline{\text{bot}}) = K \cdot P(x_*|\text{bot}). \tag{8}$$

Here $K$ simply captures the factor difference in $P(x_*)$ between benign and abuse traffic (e.g., $K = 1.1$ then $P(x_*)$ is somewhat higher in the benign traffic, if $K = 100$ it is greatly so). We emphasize that we don't assume that we know (or have good prior estimates of) $P(x_*|\overline{\text{bot}}, x_{jk})$, $P(x_*|\text{bot}, x_{jk})$ or $K$.

Comparing (3) and (6) we note that our bound differs from the actual Naive Bayes estimate by a multiplicative factor:

$$\frac{\hat{P}(x_*|\overline{\text{bot}}) - P(x_*|x_{jk})}{P(x_*|\overline{\text{bot}}) - P(x_*|x_{jk})} \cdot \frac{P(x_*|\overline{\text{bot}}) - P(x_*|\text{bot})}{\hat{P}(x_*|\overline{\text{bot}})}.$$

If we assume that at least one bin of at least one feature receives no attack traffic (i.e., $P(\text{bot}|x_{jk}) = 0$ for some $x_{jk}$) then $\hat{P}(x_*|\overline{\text{bot}}) = P(x_*|\overline{\text{bot}})$ and the amount by which we under-estimate becomes

$$\frac{P(x_*|\overline{\text{bot}}) - P(x_*|\text{bot})}{P(x_*|\overline{\text{bot}})} = 1 - 1/K.$$

That is, our estimated conditional probability is related to the actual value by:

$$\hat{P}(\text{bot}|x_{jk}) = (1 - 1/K) \cdot P(\text{bot}|x_{jk}). \tag{9}$$

Thus, when $K$ is large we get a very accurate approximation of the Naive Bayes estimate. A similar analysis yields:

$$\hat{P}(\text{bot}) = (1 - 1/K) \cdot P(\text{bot}). \tag{10}$$

If an unattacked bin does not exist the bound is still quite tight. To see this, suppose that, in the least-attacked bin, 5% of traffic is abuse. From (2) and (8) this gives $\hat{P}(x_*|\overline{\text{bot}}) = (0.95 + 0.05/K) \cdot P(x_*|\overline{\text{bot}})$. Hence we differ from the actual Naive Bayes estimate by a multiplicative factor:

$$\frac{(0.95 + 0.05/K) \cdot P(x_*|\overline{\text{bot}}) - P(x_*|x_{jk})}{P(x_*|\overline{\text{bot}}) - P(x_*|x_{jk})} \times \frac{P(x_*|\overline{\text{bot}}) - P(x_*|\text{bot})}{(0.95 + 0.05/K) \cdot P(x_*|\overline{\text{bot}})}.$$

The second term is $(1 - 1/K)/(0.95 + 0.05/K)$, which ranges between $(1 - 1/K)/0.95$ and $(1 - 1/K)$ as $K$ ranges $1 \to \infty$. The first term is close to 1 when $P(x_*|\overline{\text{bot}}) - P(x_*|x_{jk})$ is large; i.e., when the received distribution differs considerably from clean. Thus, when we do not have unattacked bins, but the volume of abuse is high, we still underestimate the Naive Bayes by $\approx (1 - 1/K)$ so long as at least one bin of at least one feature sees little abuse traffic.

### 4.2 Preservation of order

Consider now the overall estimate $P(\text{bot}|\boldsymbol{x})$. We next show that if $P(\text{bot}|\boldsymbol{x}) > P(\text{bot}|\boldsymbol{x}')$ then $\hat{P}(\text{bot}|\boldsymbol{x}) > \hat{P}(\text{bot}|\boldsymbol{x}')$ (again assuming at least one bin of at least one feature receives no attack traffic).

Observe:

$$
\begin{aligned}
\hat{P}(\text{bot}|\boldsymbol{x}) &= \frac{\hat{P}(\text{bot})}{P(\boldsymbol{x})} \cdot \hat{P}(\boldsymbol{x}|\text{bot}) \\
&= \frac{\hat{P}(\text{bot})}{P(\boldsymbol{x})} \cdot \prod_{i=1}^{d} \frac{\hat{P}(\text{bot}|x_i) \cdot P(x_i)}{\hat{P}(\text{bot})} \\
&= \frac{(1 - 1/K) \cdot P(\text{bot})}{P(\boldsymbol{x})} \cdot \prod_{i=1}^{d} \frac{(1 - 1/K) \cdot P(\text{bot}|x_i) \cdot P(x_i)}{(1 - 1/K) \cdot P(\text{bot})} \\
&= \frac{(1 - 1/K) \cdot P(\text{bot})}{P(\boldsymbol{x})} \cdot \prod_{i=1}^{d} \frac{P(\text{bot}|x_i) \cdot P(x_i)}{P(\text{bot})} \\
&= (1 - 1/K) \cdot P(\text{bot}|\boldsymbol{x}).
\end{aligned}
$$

This is true independently of $K$ (so long as $K > 1$). Thus (when we have unattacked bins) a ranking under our estimate $\hat{P}(\text{bot}|x_{jk})$ has the same order as a ranking under the true value $P(\text{bot}|x_{jk})$, even if we are wrong about the exact values. Significantly this means that the Receiver Operator Characteristic (ROC) curve that our method produces will be identical to that of the true Naive Bayes algorithm. We can use this to order browser versions, geographic regions, IP blocks, etc, by the fraction of abuse traffic they contain.

The above analysis considers the behavior in the limit of arbitrarily large amounts of data. In practice, of course, we must consider the effects of sampling. Our method behaves similarly to NB in this respect. That is, the true NB estimate is the ratio $N^+/(N^+ + N^-)$ of counts of the labels. The two terms in our estimate (6) (i.e., $P(x_*)$ and $\hat{P}(x_*|\overline{\text{bot}})$) are similarly found as ratios of counts, not of the labels but of $x_*$. Naturally, confidence intervals should be considered in the maximization calculation of (5).

### 4.3 Claims

For convenience we summarize which conclusions depend on which assumptions. Since we seek a Naive Bayes classifier we assume per-class independence of features. If, for some $x_*$, $P(x_*|\overline{\text{bot}}) > P(x_*|\text{bot})$ then

(6) gives a lower bound on the conditional probability $P(\text{bot}|x_{jk})$. If, in addition, at least one bin of at least one of the features in $\boldsymbol{x}$ receives no attack traffic then $\hat{P}(\text{bot}|x_{jk}) = P(\text{bot}|x_{jk})$. When this is true we underestimate $P(\text{bot}|x_{jk})$ by a factor of $(1 - 1/K)$, a ranking of a group of bins under our estimate will be the same as a ranking under $P(\text{bot}|x_{jk})$, and the ROC curve has the same shape as that of true NB.

### 4.4 Notes and Limitations

Whether at least one bin of at least one feature is free of attack traffic will obviously be dataset-dependent. However, if any of the features has high cardinality the assumption will be satisfied unless every bin contains attack traffic. Web traffic, for example, may be grouped by city, region, IP subnet or Autonomous System Number (ASN). If any of these features has thousands of bins, the assumption will be satisfied unless attack traffic comes from every city, region, ASN, etc.

Our analysis neglects sampling and finite-data effects. In practice, confidence intervals for all quantities should be calculated, and these may have a significant effect on the quality of the approximation, even when the main assumptions are met. The approximation of NB will thus be far better for large datasets than for small ones.

## 5 Toy example: account registration

Consider the case of account registration at a web service. Suppose (unrealistically) that benign traffic has equal volumes from all 50 states. Suppose most of the abuse traffic comes from 10 states where it makes up 30% of traffic, but there's also smaller amounts in the other 40 states where it makes up 1% of traffic.

On signup users are asked to provide an email account. Popular free domains like `gmail, hotmail` and `outlook` obviously dominate. However, a small fraction of legitimate users provide email addresses from a paid service like `AOL` or Comcast; e.g., $P(\texttt{AOL}|\overline{\text{bot}}) = 0.03$. If attackers concentrate on free domains we might expect $P(\texttt{AOL}|\overline{\text{bot}}) > P(\texttt{AOL}|\text{bot})$. Thus, we can use `AOL` as our discriminative feature $x_*$. For definiteness suppose $P(\texttt{AOL}|\text{bot}) = 0.002$ (although, of course, this information isn't available to our estimator). Observe that:

$$
\begin{aligned}
P(\texttt{AOL}|\text{state}) &= [1 - P(\text{bot}|\text{state})] \cdot P(\texttt{AOL}|\overline{\text{bot}}) + P(\text{bot}|\text{state}) \cdot P(\texttt{AOL}|\text{bot}) \\
&= [1 - P(\text{bot}|\text{state})] \cdot 0.03 + P(\text{bot}|\text{state}) \cdot 0.002.
\end{aligned}
$$

In the most-attacked states abuse is 30% of volume, so $P(\texttt{AOL}|\text{state}) = 0.7 \times 0.03 + 0.3 \times 0.002 = 0.0216$ while in the 40 less-attacked states $P(\texttt{AOL}|\text{state}) = 0.99 \times 0.03 + 0.01 \times 0.002 = 0.0297$.

Our estimate $\hat{P}(\texttt{AOL}|\overline{\text{bot}})$ in (5) will come from the least-attacked states: $\hat{P}(\texttt{AOL}|\overline{\text{bot}}) = \max_{\text{state}} P(\texttt{AOL}|\text{state}) = 0.0297$. Thus, from (9) in the states that receive most attack traffic our estimate is $1 - 0.0216/0.0297 \approx 0.273$. In the states that receive little attack traffic our estimate is $1 - 0.0297/0.0297 = 0$.

Thus, in the 10 states that account for 88% of attack traffic we under-estimate $P(\text{bot}|\text{state})$ by $0.273/0.3 = 0.91$. This is slightly worse than the $1 - 1/K = 1 - 0.002/0.03 = 0.933$ predicted in Section 4.1 because in this example there were no unattacked bins. That is we closely approximate the true Naive Bayes estimate, even though the only assumptions we make are per-class independence of features, and that free accounts are more popular with attackers than paid-for ones (e.g., the estimator does not assume knowledge of $P(x_*|\overline{\text{bot}})$, $P(x_*|\text{bot})$ or $K$).

## 6 Example: Password-spray guessing attacks

Consider the example of online password guessing. An authentication server receives login requests which are POST events containing a username and password (as well as other logged features such as useragent, timestamp, IP address, geo-location, etc). Traffic is an unknown mixture of benign (i.e., attempts by users) and malicious (i.e., guesses sent by attackers). Spray password-guessing attacks involve sending a small number of guesses (e.g., less than 100) against each of millions of different accounts. For example, the 100 most common passwords cover approximately 3% of the distribution at a large consumer platform (Bonneau,

Joseph, 2012). By trying 100 guesses against each of 1 million accounts an attacker can expect a yield of 30,000 accounts. Since no individual account receives a large number of guesses, heuristics such as locking an account after several failed attempts (i.e., "three strikes" type rules) have limited utility. Despite the prevalence of the problem there appears little beyond rules-based approaches and heuristics to address it (Florêncio et al., 2014).

Let $x_*$ be a feature that is `True` if the correct password is submitted, and `False` otherwise. We expect that legitimate users submit the correct password most of the time (failing perhaps $\approx 10\%$ or so due to forgetting, typos, etc). However, attackers must fail more than 99% of the time (since even the most common password covers <1% of accounts). Thus, $P(x_*|\overline{\text{bot}}) > 0.9$ and $P(x_*|\text{bot}) < 0.01$. Thus, conservatively, we should have $K > 90$, so that, from (9), our estimate for any particular bucket of traffic being malicious will be $1 - 1/90 \approx 0.989$ of the true conditional probability. This allows us to identify which (if any) useragents, ISPs, IP address blocks send the highest proportion of attack traffic. This in turn allows us to make finely grained decisions. Instead of blocking all requests from certain IP ranges based on rules with hard-coded thresholds (as appears common (Florêncio et al., 2014)) we can have a full Naive Bayes classifier: e.g., refusing requests only from certain IP addresses at certain times that also use certain browsers, etc.

## 7 Evaluation

We evaluate the algorithm on simulated data. We choose a random uniform $R_j$-dimensional vector for $P(x_j|\text{bot})$ (where $R_j$ is the feature cardinality along dimension $j$). For the clean distribution we choose an $R_j$-dimensional Zipf vector with exponential factor of 2. For $d$ features, with cardinalities `R[j]` this might be done in Python as follows:

```python
import numpy
for j in range(d):
    P[j] = numpy.random.zipf(2, R[j])
    Q[j] = numpy.random.rand(R[j])
    P[j] /= P[j].sum()
    Q[j] /= Q[j].sum()
```

We zero one randomly-chosen bin of one randomly chosen feature of `Q` and re-normalize. We generate a logstream by choosing a sample to be from either the clean or abuse distributions with probability $1 - P(\text{bot})$ and $P(\text{bot})$ respectively. If we choose from the clean distribution we generate the $d$ columns of a data point:

```python
for j in range(d):
    col[j] = random.choices(symb[j], P[j])
```

where `symb[j]` is an array containing the possible values of the j-th feature. Similarly, if we choose from the abuse distribution. This allows us to generate arbitrary amounts of data for testing. We showed in Section 4.2 that the ROC curve produced by our algorithm is unaffected by $P(x_*|\overline{\text{bot}})$, $P(x_*|\text{bot})$ and $K$ so long as $K > 1$. Thus, we arbitrarily chose $P(x_*|\overline{\text{bot}}) = 0.2$, $P(x_*|\text{bot}) = 0.1$ and $K = 2$; we verified that the shape of the empirical ROC curve does not depend on these values (i.e., by varying these choices). In our generated data we set $x_*$ to True with probability $P(x_*|\overline{\text{bot}})$ and $P(x_*|\text{bot})$ for each sample of the clean and abuse data respectively. Again, we emphasize that none of these choices affect the shape of the ROC curve produced by our algorithm. Our algorithm had access to $x_*$, but obviously did not have access to the clean/abuse labels, the chosen values for $P(x_*|\overline{\text{bot}}), P(x_*|\text{bot})$ or $K$. We compare our approach with true NB. The NB algorithm was trained on a labelled training set, while our Approximate NB algorithm did not have access to the labels.

Figure 1 shows our comparison ROC curves for increasing values of $P(\text{bot})$ when the clean and abuse data are chosen from Zipf and random distributions as above. Each set involved 20 million samples, $d = 4$ categorical features, each of which had cardinality $R_j = 20$. As expected, the ROC curve of our approximation matches that of NB.

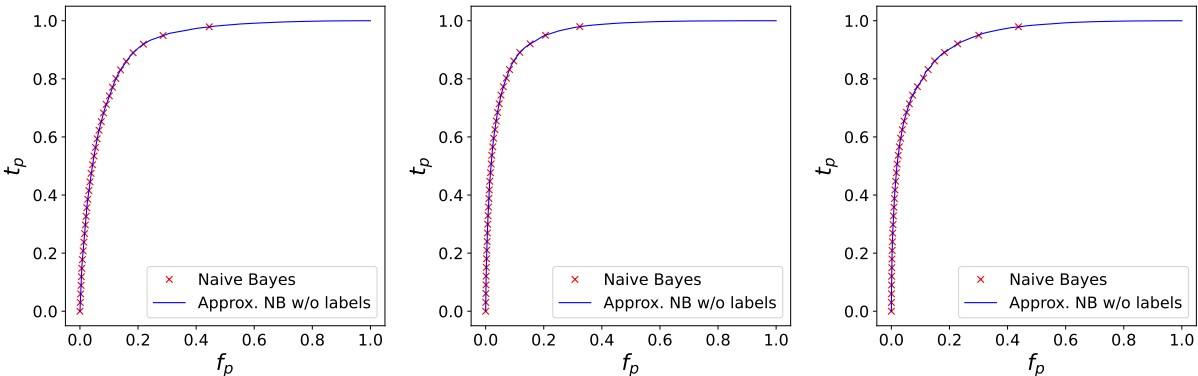

Figure 1: ROC curves comparing Naive Bayes and our approximation for synthesized data and increasing levels of abuse. The data had $d = 4$ categorical features each of cardinality 20. The clean distribution was Zipf, while the abuse distribution was uniform. (a) $P(\text{bot}) = 0.125$, (b) $P(\text{bot}) = 0.25$, (c) $P(\text{bot}) = 0.5$. Note that the Naive Bayes classifier had access to labels while our approximate approach did not.

We also test with a second dataset. Instead of Zipf the clean data, $P(x_j|\overline{\text{bot}})$, in this case is also formed from a randomly chosen $R_j$ dimensional vector. This might be done in Python as:

```python
for j in range(d):
    P[j] = numpy.random.rand(R[j])
    Q[j] = numpy.random.rand(R[j])
    P[j] /= P[j].sum()
    Q[j] /= Q[j].sum()
```

Again we zero one randomly-chosen bin of one randomly chosen feature of `Q`, re-normalize and generate a 20 million sample logstream as before. This represents a harder classification task since there is less distinction between clean and abuse distributions. Figure 2 shows the comparisons. While the classifier does worse on this task, again note that the approximate NB method closely mirrors the performance of true NB.

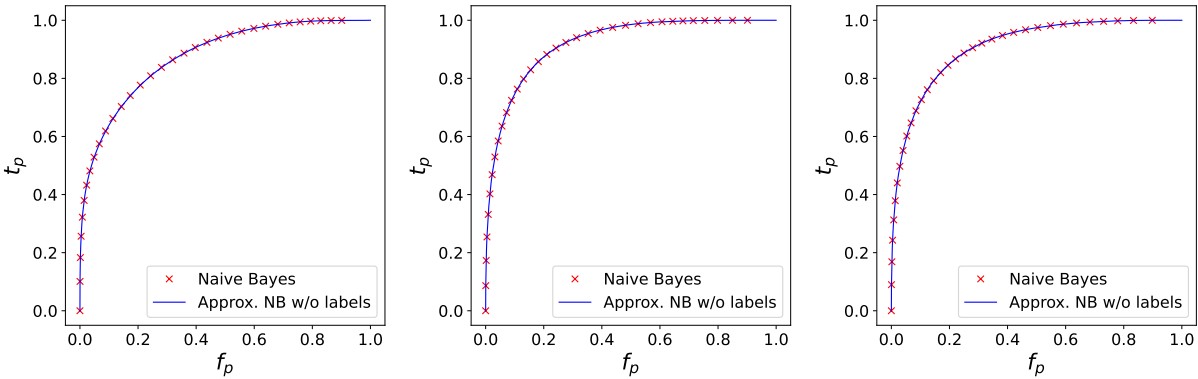

Figure 2: ROC curves comparing various methods for synthesized data and increasing levels of abuse. The data had $d = 4$ categorical features each of cardinality 20. Both clean and abuse distributions were uniform. (a) $P(\text{bot}) = 0.125$, (b) $P(\text{bot}) = 0.25$, (c) $P(\text{bot}) = 0.5$. Note that the Naive Bayes classifier had access to labels while our approximate approach did not.

## 8    Discussion

We've given a method to approximate an NB classifier. We rely on being able to identify a binary feature $x_*$ that is more prevalent in the benign than in the abuse traffic and the Naive Bayes assumption of per-class independence of the features.

Our approach is conservative in that we underestimate the odds that traffic is abuse. Under the assumption that at least one bin of at least one feature receives no attack traffic: when $K$ is large, our estimate will be closely approximate the true Naive Bayes conditional probability. The ROC curve of our method matches that of true NB (except for sampling effects when data is limited).

We do not claim that this produces excellent classification decisions or that it will be state-of-the-art for any particular application. Rather, it provides a new possibility when the alternatives are very limited. This is often the case in abuse problems where there can be high volumes of abuse and there is no possibility of labels. When this is so, a feature satisfying (8) can act as a very useful weak supervision signal. It is also useful in allowing us to rank subsets of the data by the estimated amount of abuse traffic.

## 9    Conclusion

We have shown an approach that allows us to approximate the Naive Bayes estimator when we have categorical features and no labels are available. We require a feature $x_*$ that is more prevalent in the legitimate than the scripted traffic.

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
