# OpenReview forum: "Approximating Naive Bayes on Unlabelled Categorical Data"
_TMLR — Accepted by TMLR_

### Review · Reviewer_YFuT · 2023-05-02

**Summary Of Contributions:**

The paper proposes a theoretically-principled method for approximating Naive Bayes on unlabeled datasets with categorical features.  The key assumption is there exists a binary feature that is more prevalent in one class than in the other.  The relevant prevalence controls the acuracy of approximation.  Under further assumptions, the method preserves the ordering of data points, and thus produces the same ROC curve as Naive Bayes.

**Audience:**

Yes

**Broader Impact Concerns:**

I don't have concerns.

**Claims And Evidence:**

Yes

**Requested Changes:**

(Also including detailed comments here.  The authors can use their judgment regarding the more subjective suggestions.)

Last sentence on page 1, "Varol et al"  (and later in the paper): missing "."; consider using \citeauthor?

"Legitimate users use paid-for email services": I'm not sure to what extent this is still true --- any evidence?

Related work, one-hot encoding: in principle one can define the distance in richer ways.  For example, Win98 is probably closer to WinXP than to MacOS 10.0.

Second paragraph of sec 3: "discus".

Next paragraph, "replace each token with the conditional probability": in general information is lost in this process, right?  Any theoretical justification of this approach?  (Well I see that very soon you mention the independence assumption...)

Paragraph after eq (4), "(4) implies ...": I guess what you really mean is P(x_* | bot) \ge 0 (and the inequality doesn't need to be strict)...

Sec 4.3: "at least one bin is free of attack": this is a bit tricky when considered together with confidence intervals.  As the authors argue, it is quite plausible that no attack comes from some city / region / ASN, but then taking into consideration confidence intervals one can't just say the true probability of an attack is 0.  In fact there's a tradeoff between "seeing no attack in the dataset" and "being sure that the true probability isn't too much higher", and I think it's somewhat unfair to emphasize only the former.

**Strengths And Weaknesses:**

Strengths

The method is simple and reasonable, and the theoretical guarantees are quite good given what is assumed.  As the authors argue, the method might be practically useful in settings where no other method easily applies.


Weaknesses

I think an empirical evaluation of the method would make the paper much stronger, and better support the claims.  Also see detailed comments below for some minor issues.

---

> ### Author Response · Authors · 2023-05-02
> **Thanks**
>
> Thankyou for your review.
>
> >I think an empirical evaluation of the method would make the paper much stronger, and better support the claims.
>
> Obviously we agree that evaluation on real-world data would add a lot. However, we've been unable to identify a suitable dataset. Organizations appear particularly shy about releasing data for security applications. In the password-guessing example we'd need not just fields like useragent, etc but also the submitted password. We're pretty confident that no such dataset is publicly available. Also, the lack of labels (which we'd need for metrics like AUC etc) is pretty inherent to some problems, and not just a question of cost or resources. Eg, there's no feasible way of having a human grader look at advertising web-logs and decide which clicks are real and which are bot.
>
> >Next paragraph, "replace each token with the conditional probability": in general information is lost in this process, right? Any theoretical justification of this approach?
>
> Yes, this step is aggregating, so if we had labels most of the information they contain is lost here. The only theoretical justifications we've seen for this are as a stepping stone to Naive Bayes (eg, Bilenko and Zheng and Casari).
>
> >Sec 4.3: "at least one bin is free of attack": this is a bit tricky when considered together with confidence intervals.
>
> Yes, "haven't seen" and probability of attack equal to zero are different things. In the limit of very large data the problem will be small, but is more pronounced at smaller scales. We think it might be useful to add some text to 4.3 about the importance of allowing confidence intervals around all estimated quantities, and pointing out that this approach will be harder to apply at small scale than at large. Thanks for suggesting.

---

> ### Author Response · Authors · 2023-05-04
> **Forgot to address this point**
>
> >Related work, one-hot encoding: in principle one can define the distance in richer ways. For example, Win98 is probably closer to WinXP than to MacOS 10.0.
>
> Absolutely, these are close in some human-interpretable sense. But we need a distance that implies greater likelihood of belonging to the same class. That is, even if an attacker is sending lots of traffic from WinXP this doesn't tell us anything about Win98, since we can't assume P(bot|Win98) ~ P(bot|WinXP). It's perfectly possible that P(bot|Chrome78) ~ 1 and P(bot|Chrome79) ~ 0.

---

### Review · Reviewer_cWPS · 2023-05-03

**Summary Of Contributions:**

This paper proposes an Approximated Naive Bayes (NB) classifier targeting for unlabeled and categorical data points. It relies on the assumption that we can have a feature x* has higher chance in the benign setting over abused ones. Also, another assumption is per-class independence of the features. The paper find good motivating settings for the real use cases of this proposed NB classifier.

**Audience:**

Yes

**Broader Impact Concerns:**

No ethical concerns

**Claims And Evidence:**

Yes

**Requested Changes:**

See weakness above for evaluation comments.

In terms of writing, I would suggest having a subsection in related work, mentioning (1) in general how people dealing with related abnormal detection in existing works, and (2) how the investigated problem scenarios in this paper differ from existing ones.

**Strengths And Weaknesses:**

### Strength
1. The proposed idea is quite interesting and seems promising under specific real scenarios (unlabeled data, categorical data). The paper highlight several practical settings that well fit the limitations of the assumptions (existing feature with a higher benign chance + independence).
2. Two toy examples are given to show fairly good approximations by the proposed NB estimator and both example settings seem quite practical.

### Weakness
1. The proposed approach targets for specific scenarios due to assumption constraints.
2. Additional empirical evaluation would help a lot to the paper. As mentioned in the related work that existing unsupervised classifier with one-hot encoding and one-class SVM could likely perform poorly under unlabeled categorical settings. However, it would be very helpful to have some existing common methods evaluated in the provided toy examples as well to confirm this point. In specific the paper claims the distance between points and the fraction of samples expected to be anomalous are the main limitation factors. It would be interesting to see, comparing the proposed method to K-means and One-class SVM, how the estimation tightness changes in terms of changing these factors. This would be very helpful to guide real users when to use the proposed NB classifier over the others.
3. It would be great to show empirical ablation study of estimation tightness in terms of states and K for one toy example with specific feature like AOL.

---

### Review · Reviewer_2CVy · 2023-05-11

**Summary Of Contributions:**

The idea behind this work is that, if we are willing to accept a model as naive as naive bayes, we can exploit prior information about the conditional probability of the class label given specific features to approximate the naive bayes solution. This is valuable in cases where the labeling of a non-trivial amount of data is very hard, as is often the case in information security problems.

**Audience:**

No

**Claims And Evidence:**

No

**Requested Changes:**

 In particular, the obvious ablation seems to be:

1. Train Naive Bayes on the ground truth labels $y$
2. Train this approach on some 1-5 features that are most biased toward $y$, but hold back the true $y$ labels.
3. Train naive bayes using $\hat{y} = x_*$ to quantify the empirical benefit of using this approach over assuming that the feature should instead be used as a proxy label.

There are also some related works from the information security space that seem critically relevant given the approach in the paper. Most notable is the Firenze approach presented at CAMLIS 2022 ( https://arxiv.org/abs/2207.00827 )  and the Approximate Ground Truth Refinement (AGTR) from AISec 2021 ( https://arxiv.org/abs/2109.11126 ) . Both of these works deal with the evaluation of malware classified on unlabeled data by using some form of side or prior information. While different from the proposed work, they are more directly relevant than the cited literature as those in the IS community have worked on tackling these issues already.

While the AGTR should be noted in related work, I do think it is appropriate to perform a comparison against a (mildly adapted) Firenze, as it is employing the same key insight: some features are known apriori to occur more frequently in $y$ and in $\neg y$, and that can be achieved with a small number of such features. Firenze also provides the foundation for this exploration via the Ember dataset.

I think this would make the work complete and demonstrate that the proposal is worth consideration and use, and fully support the claim that this is a viable approach to leverage.

**Strengths And Weaknesses:**

The idea behind the paper is clever, and gets at a niche - but very real - problem in the information security space. The theory is simple and makes sense in how they approached the problem and potentially gives life and utility to the often inferior naive Bayes.

While very desirable, I think the most significant item missing from this work is some empirical evaluation. Even if industry data can not be obtained (e.g., the password attack example), many datasets exist and could be used to explore the efficacy of this approach. Given the very empirical motivation of the paper, it is thus hard to tell if the approach is actually of utility. This is notable because the idea behind the paper has been explored in other works in the information security realm, and the strategy can be leveraged at large without relying on the specific approach given (i.e., use noisy label sources and train models assuming label noise occurs).

-----

To avoid confusion, the author's current stance is that the paper is only claiming math and application. In this light, I think the paper as a whole needs major re-writing as the examples, notation, and motivation all lean heavily to imply a practical use in information security. However, this renders the paper - in my opinion and reading of the guidelines - uninteresting to the TMLR audience. Given the guidelines state that if there is any doubt, one should assume it would be applicable: I am confident that it would not be of interest. I hope this clears up confusion on this reviewer's stance on what actions can be taken to improve the manuscript. I do earnestly hope it is headed as there is a niche idea that could prove insightful, but I don't believe any insight can be drawn without some empirical results on real data.

---

> ### Author Response · Authors · 2023-05-11
> **Thankyou, and which claims are not supported?**
>
> Thankyou for your review.
>
> Could you be explicit as to which claims in our submission you found to be not supported? As a reminder, you can take the first paragaph of Section 4.3 as the definitive statement of what we are claiming.

---

> > ### Comment · Reviewer_2CVy · 2023-05-11
> > **Not supported claims**
> >
> > >we offer a better-than-nothing classifier
> >
> > The two alternatives in the abstract, are used in practice today. For the proposed method to maybe be more useful, it needs some empirical demonstration (my review & our discussion provided ablations and approaches to achieving this).
> >
> > > The approach is particular suited to detecting guessing attacks, where K tends to be high (and thus the approximation is very accurate). Example problems (and relevant attributes) are: password-guessing, where we expect login attempts from legitimate users will succeed at a much higher rate than those from password-guessing attackers; Credit Card Verification Value (CVV) guessing, where an attacker exhaustively tries all possible 3 or 4-digit values; account registration, where we expect legitimate users will use email addresses from paid-for services at a much higher rate than attackers; click-fraud where we expect legitimate users to visit pages and services that contain no ads at a higher rate than click-fraud bots
> >
> > These are all empirical claims with no empirical support. The utility of this is predicated on the unstated assumption that
> >
> > 1. Naive Bayes is _good enough to be useful_ in the final application, and
> > 2. Better than possible ablations previously mentioned.
> >
> > >However, we point out that in many applications some features may have very high cardinality. Web traffic, for example, may be grouped by city, region, IP subnet or Autonomous System Number (ASN), each of which might have thousands of bins. Unless attack traffic comes from every city, region, ASN, etc, the assumption will be satisfied.
> >
> > This is also an empirical statement.

---

> > > ### Author Response · Authors · 2023-05-12
> > > **Thankyou for the clarification**
> > >
> > > >>we offer a better-than-nothing classifier
> > >
> > > >The two alternatives in the abstract, are used in practice today. For the proposed method to maybe be more useful, it needs some empirical demonstration (my review & our discussion provided ablations and approaches to achieving this).
> > >
> > > We have reworded to describe the approach as an alternative to a) doing nothing and b) heuristics rather than stating or implying that it is better than them.
> > >
> > > >>The approach is particular suited to detecting guessing attacks, where K tends to be high (and thus the approximation is very accurate). Example problems (and relevant attributes) are: password-guessing, where we expect login attempts from legitimate users will succeed at a much higher rate than those from password-guessing attackers; Credit Card Verification Value (CVV) guessing, where an attacker exhaustively tries all possible 3 or 4-digit values; account registration, where we expect legitimate users will use email addresses from paid-for services at a much higher rate than attackers; click-fraud where we expect legitimate users to visit pages and services that contain no ads at a higher rate than click-fraud bots
> > >
> > > >These are all empirical claims with no empirical support. The utility of this is predicated on the unstated assumption that
> > > >Naive Bayes is good enough to be useful in the final application, and
> > > >Better than possible ablations previously mentioned.
> > >
> > > We don't see that basic statements like these should be interpreted as claims (eg, that a person who is trying to guess the password will fail at a higher rate than someone who knows it). Nonetheless we have reworded: instead of offering examples of the form 'Problem Y where we expect condition X' we now say 'Problem Y if X.'
> > >
> > > We are not claiming that Naive Bayes is good enough for any particular application.
> > > Having removed the 'better-than-nothing' phrase we are not claiming that this is better than a) doing nothing or b) heuristics. The claim is simply that it offers an alternative to a) and b).
> > >
> > > >>However, we point out that in many applications some features may have very high cardinality. Web traffic, for example, may be grouped by city, region, IP subnet or Autonomous System Number (ASN), each of which might have thousands of bins. Unless attack traffic comes from every city, region, ASN, etc, the assumption will be satisfied.
> > >
> > > >This is also an empirical statement.
> > >
> > > We disagree that every non-mathematical statement should be considered a claim. The conclusion in the last sentence is explicitly conditioned on the assumptions in the preceding sentences.
> > >
> > > Nonetheless we have reworded to make the conditional nature clearer. We've removed statements about what 'may' and 'might be' and again written in the form 'Conclusion Y if condition X.'
> > >
> > > We're uploading a new version with these changes. Please let us know if these changes satisfy you that the claims are now supported, or if you have other concerns.

---

> > > > ### Comment · Reviewer_2CVy · 2023-05-12
> > > > **Empirical Results Required**
> > > >
> > > > As I've stated before, I do not believe you can reduce the scope of your claims and keep the requirement of TMLR that the submission be of interest to at least some TMLR reviewers. The mathematics and theory in this work is insufficient in its own right to be of interest as a purely theoretical work. The idea of how this could be useful is interesting, but _requires_ some empirical results to demonstrate. This is an implicit statement of the entire paper - e.g., the use of $\neg$bot and bot as classes presupposes the utility of this in such an application. But no actual evidence of this method's utility is presented, and I do not believe utility, in this case, can be demonstrated outside empirical results. As my review highlighted, there is already work in this same vein from the IS community that has the same fundamental insight but also includes empirical results. I've also detailed how you can add empirical results to this paper, in a way that requires no special knowledge beyond the ability to read/write Python code.
> > > >
> > > > If reducing the claim scope is something I felt would help convince me to vote accept on your paper, I would do so. I do not believe this is an option for your paper, following the TMLR guidelines.
> > > >
> > > > I will add, in a previous reply you said you agreed and would update the related work based on the Firenze and AGTR papers. Not doing so, and instead pursuing edits that you yourself said were not an avenue of discussion you wanted to continue, is poor form.
> > > >
> > > > Beyond the scholastic issues, if empirical results are added - of which how to obtain a dataset and convert it to binary features has been provided to you - and properly ablated, I'm happy to vote for acceptance. However, if the authors believe that utility can be separated from empirical results, it is the author's job to provide a convincing argument as such. Conditioning every statement of utility or assuming utility is not convincing to me at this time.

---

> > > > > ### Author Response · Authors · 2023-05-12
> > > > >
> > > > > >As I've stated before, I do not believe you can reduce the scope of your claims and keep the requirement of TMLR that the submission be of interest to at least some TMLR reviewers. The mathematics and theory in this work is insufficient in its own right to be of interest as a purely theoretical work.
> > > > >
> > > > > Yes, we understood from the previous discussion that this is your view. However, your review indicated that Yes the claims were of interest to the audience, but No they were not supported. Since the claims are mathematical it would seem reversing those answers would make sense. Empirical results might be required, in your view, to make the claims interesting, but not to support them.
> > > > >
> > > > > So we repeat the question: can you state explicitly which claims of the submission you find to be not supported?
> > > > >
> > > > > > I've also detailed how you can add empirical results to this paper, in a way that requires no special knowledge beyond the ability to read/write Python code.
> > > > >
> > > > > As we explained on the previous thread: the data you pointed to does not have categorical features, none of the authors have any experience with malware analysis, and we suspect the mapping to categorical features that you suggest require domain knowledge.
> > > > >
> > > > > We point out that the TMLR acceptance criteria explicitly endorse reduction of claims as an alternative to reviewer-requested experiments: https://jmlr.org/tmlr/acceptance-criteria.html
> > > > >
> > > > > >Any gap between claims and evidence should be addressed by the authors. Often, this will lead reviewers to ask the authors to provide more evidence by running more experiments. However, this is not the only way to address such concerns. Another is simply for the authors to adjust (reduce) their claims.
> > > > >
> > > > > >I will add, in a previous reply you said you agreed and would update the related work based on the Firenze and AGTR papers. Not doing so, and instead pursuing edits that you yourself said were not an avenue of discussion you wanted to continue, is poor form.
> > > > >
> > > > > We uploaded a revision as a courtesy so that you might immediately check our rewording of parts that you found problematic. This was not intended as final. We have not got around to adding references to the Firenze and AGTR papers or indeed fixing some things pointed out by the other two reviewers. Please assume good intent.

---

> > > > > > ### Comment · Reviewer_2CVy · 2023-05-13
> > > > > >
> > > > > > >Since the claims are mathematical it would seem reversing those answers would make sense.
> > > > > >
> > > > > > As I stated in the message before this one, "This is an implicit statement of the entire paper...", the entire paper is written in a manner that pre-suposes utility in an empirical setting. I do not believe you can reduce your claim set for the paper without basically deleting the motivation and re-writing all example to avoid using bots/passwords etc as examples, because there is no empirical evidence that you can actually do this on real data.
> > > > > >
> > > > > > If you were to do such an edit, I would vote that the paper is of no relevance to any sub-population of TMLR.
> > > > > >
> > > > > > >Empirical results might be required, in your view, to make the claims interesting, but not to support them.
> > > > > >
> > > > > > No, I said explicitly that I believe empirical results are mandatory for the nature of how this paper is written, and that a re-write avoiding ll discussion of the supposed utility /potential in a IS context would render the paper uninteresting.
> > > > > >
> > > > > > >As we explained on the previous thread: the data you pointed to does not have categorical features, none of the authors have any experience with malware analysis, and we suspect the mapping to categorical features that you suggest requires domain knowledge.
> > > > > >
> > > > > > And I replied with details on how you can convert the Ember dataset to be one that has only categorical features by going through the code, and taking each group of feature-hashed variables by type, and using PCA/SVD to create binary features.
> > > > > >
> > > > > > I also gave you a second option to improve the paper, which while I think insufficient would be far improved, to use b-lean/other tools to construct Bayesian networks to have a synthetic ground truth to demonstrate results.
> > > > > >
> > > > > > Your lack of background is not a reason for a reviewer to accept a paper.
> > > > > >
> > > > > > >We point out that the TMLR acceptance criteria explicitly endorse the reduction of claims as an alternative to reviewer-requested experiments: https://jmlr.org/tmlr/acceptance-criteria.html
> > > > > >
> > > > > > I am aware, and I already stated many times I don't think you can reduce your claim set, and in the very message you are replying to you I stated I would have suggested it if I felt it would be satisficing for this work. I don't believe it is an option for this paper.
> > > > > >
> > > > > > >Please assume good intent
> > > > > >
> > > > > > I've tried very hard to provide instructions on how more empirical results could be added. Every reviewer has mentioned this as a weakness of the paper, and I tried to provide significant instruction on how to use the Ember dataset as the most accessible, how you could support your arguments to a lesser degree with synthetic data, and I believe have made my stance that the reduced claim set is not sufficient early on. The continued focus on reducing the claim set comes across as disingenuous to an interest in improving the fundamental weakness of the paper.
> > > > > >
> > > > > > I also mentioned before the authors are free to disagree with me and appeal to other reviewers/editor. The reviewers are also free to try and convince me that I'm wrong and a change in scope would be relevant to some TMLR populations, but appealing to the guidelines that it is _allowed_ is not a convincing argument that it is _applicable_.

---

### Comment · Reviewer_2CVy · 2023-04-23
**Interesting but not yet complete**

The idea behind this work is that, if we are willing to accept a model as naive as naive bayes, we can exploit prior information about the conditional probability of the class label given specific features to approximate the naive bayes solution. This is valuable in cases where the labeling of a non-trivial amount of data is very hard, as is often the case in information security problems.

While very desirable, I think the most significant item missing from this work is some empirical evaluation. Even if industry data can not be obtained (e.g., the password attack example), many datasets exist and could be used to explore the efficacy of this approach. In particular, the obvious ablation seems to be:

1. Train Naive Bayes on the ground truth labels $y$
2. Train this approach on some 1-5 features that are most biased toward $y$, but hold back the true $y$ labels.
3. Train naive bayes using $\hat{y} = x_*$ to quantify the empirical benefit of using this approach over assuming that the feature should instead be used as a proxy label.

There are also some related works from the information security space that seem critically relevant given the approach in the paper. Most notable is the Firenze approach presented at CAMLIS 2022 ( https://arxiv.org/abs/2207.00827 )  and the Approximate Ground Truth Refinement (AGTR) from AISec 2021 ( https://arxiv.org/abs/2109.11126 ) . Both of these works deal with the evaluation of malware classified on unlabeled data by using some form of side or prior information. While different from the proposed work, they are more directly relevant than the cited literature as those in the IS community have worked on tackling these issues already.

While the AGTR should be noted in related work, I do think it is appropriate to perform a comparison against a (mildly adapted) Firenze, as it is employing the same key insight: some features are known apriori to occur more frequently in $y$ and in $\neg y$, and that can be achieved with a small number of such features. Firenze also provides the foundation for this exploration via the Ember dataset.

I think this would make the work complete and demonstrate that the proposal is worth consideration and use, and fully support the claim that this is a viable approach to leverage.

---

### Comment · Action_Editors · 2023-05-11
**Rolling discussion starts**

Dear Authors,

You will have two weeks to rebuttal and discuss with the reviewers, who will submit their final recommendations in two weeks.

Best wishes,
AE

---

> ### Author Response · Authors · 2023-05-19
>
>  We have uploaded a revised version.
>
> We agree without reservation with all three reviewers that this paper would be greatly improved by empirical evaluation on a real-world dataset. However, we have been unable to identify any that are suitable (and have searched extensively). Releasing server logs that have security implications is extremely rare, and the existence of the ground-truth labels needed for evaluation is even more unlikely.
>
> While we appreciate Reviewer 2CVy's suggestion, the Ember malware dataset is not suitable:\
> 	-- It does not have categorical features. This is not a minor detail: the restriction to categorical features is highlighted in the title of the submission.\
> 	-- None of the authors of this paper have any experience whatever with malware detection.\
> 	-- Mapping of features to categorical does not seem the trivial task that Reviewer 2CVy suggests. Any such mapping will lose much of the information in the features. Doing so while achieving good performance will almost certainly require domain knowledge.
>
> While we would prefer to present results on real-world data we point out that the TMLR acceptance criteria explicitly suggest that an alternative to requested experiments 'is simply for the authors to adjust (reduce) their claims.'
>
> We point out that Reviewer 2CVy's insistence on empirical validation of implicit claims that they identify seems narrower than the TMLR stated goal to 'facilitate scientific discourse on topics that are deemed less significant by contemporaries but may be important in the future.'
>
>
> Reviewer YFuT:\
> In addition to our comments on May 2: We've fixed the minor text matters. We changed the example about paid-for email services to ones where it is not easy to set up anonymous free accounts as this probably represents a much larger population. We added a comment in Section 4.4 on the importance of error analysis when dealing with finite data.
>
>
> Reviewer cWPS:\
> In addition to our comments on May 4: We added subsections to Related work as requested. We now review unsupervised and weakly supervised learning approaches, and have a separate section discussing approaches to detecting abusive bot traffic. We added several references in both subsections.
>
> Reviewer 2CVy: \
> Efforts to get reviewer 2CVy to explain which claims they believe to be unsupported or to discuss how the claims might be reduced have not gone well. Their response to our last attempt (here: https://openreview.net/forum?id=KpElM2S9pw&noteId=9FwHGEnQvB) appears to argue that mentioning password-guessing and bots as motivating examples constitute implicit claims of utility. We disagree. Motivating examples are not generally considered claims; neither are basic statements like 'some features may have very high cardinality' (from Reviewer 2CVy comment https://openreview.net/forum?id=KpElM2S9pw&noteId=RkBOCaL3AJ).
>
> We respect Reviewer 2CVy's right to the view that the theoretical contribution of this paper 'is of no relevance to any sub-population of TMLR' (https://openreview.net/forum?id=KpElM2S9pw&noteId=9FwHGEnQvB). We regret, however, that we were unable to persuade them that the claims are only those listed in Section 4.3, and that they continue to insist that our claims are unsupported.

---

> > ### Comment · Reviewer_2CVy · 2023-05-19
> > **Evidence Speaks louder**
> >
> > Simply stating that:
> >
> > >Any such mapping will lose much of the information in the features. Doing so while achieving good performance will almost certainly require domain knowledge.
> >
> > Is unconvincing, as the goal isn't to achieve SOTA, as I said in another reply, and is a part of TMLR's guidelines. The goal is to provide evidence of the efficacy of your approach, *relative to alternative ablations* as suggested. Especially in light that I had provided a reference of Firezen that uses binary features in a different way with demonstrated efficacy, so the authors do not even need to come up with their own ideas - they can use the ones already developed for them in a prior paper.
> >
> > In addition, I provided you at a minimum approach to generate data under different scenarios using blearn/any bayesian network tool, which you have ignored, but would allow you to at least perform ablations and demonstrate a simulation study.
> >
> > >None of the authors of this paper have any experience whatever with malware detection.
> >
> > The authors keep bringing this up, when the paper they've written is framed entirely in an IS context. Perhaps the authors should go find a co-author who has some IS experience who can help them test this.